# The coverage of maternal continuum-of-care and associated factors in the Lao People's Democratic Republic: A population-based cross-sectional study

Sengdavy Xaypadith[1,2], Ai Aoki[1,3], Kimihiro Nishino[1], Eiko Yamamoto[1*]

**1** Department of Healthcare Administration, Nagoya University Graduate School of Medicine, Nagoya, Aichi, Japan, **2** Department of Health Personnel, Ministry of Health, Vientiane Capital, Lao PDR, **3** Asian Satellite Campuses Institution, Nagoya University, Nagoya, Aichi, Japan

* yamamoto.eiko.f3@f.mail.nagoya-u.ac.jp

## Abstract

Maternal, newborn, and child health remains a global challenge, with continuum-of-care serving as a key strategy to improve health outcomes. This study aimed to examine the coverage of maternal continuum-of-care and to identify factors associated with continued care in the Lao People's Democratic Republic (Lao PDR). This is a cross-sectional study including 2,612 women aged 15–49 years who participated in Lao Social Indicator Survey III, 2023, gave live birth in the two years preceding the interview, and responded to questions about antenatal care (ANC), postnatal care (PNC), and delivery assistance during their last pregnancy and childbirth. Maternal continuum-of-care was defined as having ≥ 4 ANC visits, delivery by a skilled birth attendant (SBA), and at least one PNC visit after discharge and within six weeks postpartum. In the selection process of the study participants, all women who had home delivery were excluded due to missing data on the three components of the continuum-of-care. Of the 2,612 women (with facility-based delivery), 83.1% received ≥ 4 ANC visits, 82.2% received both ≥ 4 ANC visits and SBA delivery, and 3.3% completed the maternal continuum-of-care. Factors associated with completion of the continuum-of-care included classifying as a high wealth index category (adjusted odds ratio [AOR] = 2.19, 95% confidence interval [CI]: 1.19–4.04), having the last child as male (AOR = 1.65, 95% CI: 1.04–2.61) and receiving a maternal health check before discharge from the health facility (AOR = 2.40, 95% CI: 1.06–5.41). Hmong-Mien women were significantly less likely to complete the continuum-of-care than Lao-Tai women (AOR = 0.24, 95% CI: 0.06–0.91). The completion of the maternal continuum-of-care in Lao PDR was very low. Strengthening maternal health checks before discharge, addressing gender norms and financial barriers, and promoting culturally sensitive, community-based, and family-engaged approaches may improve coverage.

**Data availability statement:** The minimal data set underlying the findings of this study is available in figshare (DOI: https://doi.org/10.6084/m9.figshare.31145893). The original LSIS III data are available from the UNICEF's Multiple Indicator Cluster Surveys website repository (https://mics.unicef.org/surveys) upon request and approval.

**Funding:** The author(s) received no specific funding for this work.

**Competing interests:** The authors have declared that no competing interests exist.

## Introduction

The continuum-of-care for maternal, newborn, and child health (MNCH) refers to the integrated and uninterrupted provision of services throughout pregnancy, childbirth, and the postpartum period, including antenatal care (ANC), intrapartum care, and postnatal care (PNC) [1–3]. An integrated system to deliver the continuum-of-care for MNCH is essential to reduce morbidity and mortality among mothers, newborns, and children [1]. The World Health Organization (WHO) recommends at least eight ANC visits to promote women's health during pregnancy [1,4]. For intrapartum care, deliveries assisted by skilled birth attendants (SBA) at health facilities are recommended to prevent maternal deaths [5,6], and timely PNC visits after discharge from health facilities to detect and manage postpartum complications [1,7]. A systematic review showed that completing the continuum-of-care significantly reduces neonatal mortality, decreasing the risk of neonatal death by 16% [2].

Although substantial progress has been made in reducing maternal and child mortality globally, many low- and middle-income countries (LMICs) continue to experience preventable maternal and child deaths, and the pace of improvement remains insufficient [8–10]. Persistent gaps in access to and continuity of MNCH services, particularly during the postpartum period, contribute to these outcomes. In this context, improving the coverage and continuity of essential MNCH services has become a global priority. Reflecting this urgency, the Sustainable Development Goals (SDGs) set targets to reduce the global maternal mortality ratio (MMR) to fewer than 70 per 100,000 live births and the under-five mortality rate to fewer than 25 per 1,000 live births by 2030 [8–10].

Globally, the coverage of the continuum-of-care varies substantially across countries, particularly in relation to their level of socioeconomic development. In high-income countries, coverage across all three stages of the continuum-of-care is high, approaching 100%. In LMICs, coverage remains low, especially in sub-Saharan Africa and parts of South Asia [11–19]. A systematic review analyzing data from 17 sub-Saharan African countries found that the continuum-of-care for MNCH declined at each successive stage, with most women receiving ANC and intrapartum care, but not PNC [20].

The Lao People's Democratic Republic (Lao PDR) is a lower middle-income country in Southeast Asia, with a population of 7,769,819 in 2024 [21]. In 2023, 67.6% of the population lived in rural areas, and 5.4% lived in rural areas without road access [22]. The MMR in Lao PDR declined substantially, from 579 per 100,000 live births in 2000 to 126 in 2020 [8]. During the same period, coverage of at least four ANC visits and childbirth attended by an SBA increased [5,13,22–24]. These improvements can be attributed to the Lao government's efforts to achieve the 2015 Millennium Development Goals and ongoing initiatives targeting the SDGs. The Free MNCH Policy was introduced in 2013 to improve access to MNCH services [25]. Concurrently, the health workforce was expanded, with particular emphasis on increasing the number of SBAs in rural areas [26–29]. The National Strategy and Action Plan for Reproductive, Maternal, Newborn, Child, and Adolescent Health (RMNCAH) 2016–2025

was developed to promote service integration, quality improvement, and community engagement [26,27,30]. The national practical guidelines for ANC and PNC were revised to align with WHO recommendations and incorporated into health worker training [31]. In 2019, the Quality Assessment and Improvement Support program was launched to enhance service quality by monitoring care, identifying challenges, and supporting continuous improvements in RMNCAH services [26,27]. However, the MMR and child mortality rates in Lao PDR remain higher than those in other Southeast Asian countries [5,23,24,32]. Regarding continuum-of-care for MNCH, the coverage was reported to be 6.8% in Khammouane Province in 2016 [33], but no nationwide study has assessed MNCH continuum coverage. Therefore, this study aimed to evaluate the coverage of the maternal continuum-of-care and to identify associated factors among Lao women using nationally representative survey data.

## Material and methods

### Study design

This cross-sectional study is a secondary data analysis of the Lao Social Indicator Survey (LSIS) III [22]. The data was obtained from the Multiple Indicator Cluster Surveys website repository (https://mics.unicef.org/surveys).

Data collection of the LSIS III took place between March and August 2023 and a systematic sampling approach was employed to ensure national representativeness. The main sampling strata were defined as urban and rural areas across 18 provinces. Household selection was carried out in two stages. In the first stage, a predetermined number of census enumeration areas (1,050 sample villages) was selected using a systematic probability proportional to size sampling technique. Twenty households were systematically chosen from each selected area for the second stage (total 21,000 households). A total of 20,325 households were interviewed, and 22,512 women aged 15–49 years residing in these households were identified. Ultimately, data were collected from 22,126 women [22].

### Study participants

The inclusion criteria of this study were women aged 15–49 years who participated in LSIS III and had given live birth within the two years preceding the interview. The exclusion criteria were women with missing data on key outcome variables, namely the number of ANC visits, delivery attended by an SBA, or the number of PNC visits. Sampling weights were applied in this study except for the total number of LSIS III participants. Of all the women who participated in LSIS III, 3,448 had given live birth in the previous two years (Fig 1). However, 836 of them had missing data on the number of ANC visits and receipt of PNC after discharge. As a result, 2,612 women were included in the final analysis. The analytical sample was restricted to women who reported facility-based delivery, defined as delivery at a hospital, health center, or private clinic. Women with missing information on ANC visits, delivery assistant, or PNC were excluded from the analysis; consequently, women who delivered at home were not included in the study population.

### Socio-demographic characteristics

Socio-demographic variables included age, educational level, ethnicity, residential region, type of living area, wealth index quintile, marital status, age at first marriage, and health insurance coverage. Women's age was grouped into four categories: 15–19 years, 20–29 years, 30–39 years, and 40–49 years. Educational level was categorized as "none/elementary," "secondary" (combining lower and upper secondary), and "post-secondary/tertiary." Ethnicity was classified into five groups: Lao-Tai, Mon-Khmer, Hmong-Mien, Chinese-Tibetan, and other. Provinces of residence were categorized into northern, central, and southern regions. Living area was classified as either urban or rural. Wealth index quintiles (poorest, second, middle, fourth, and richest) were grouped into two categories: "low/middle (poorest, second, and middle)" and "high (fourth and richest)." Marital status was categorized as married (formally married or living with a partner) and not

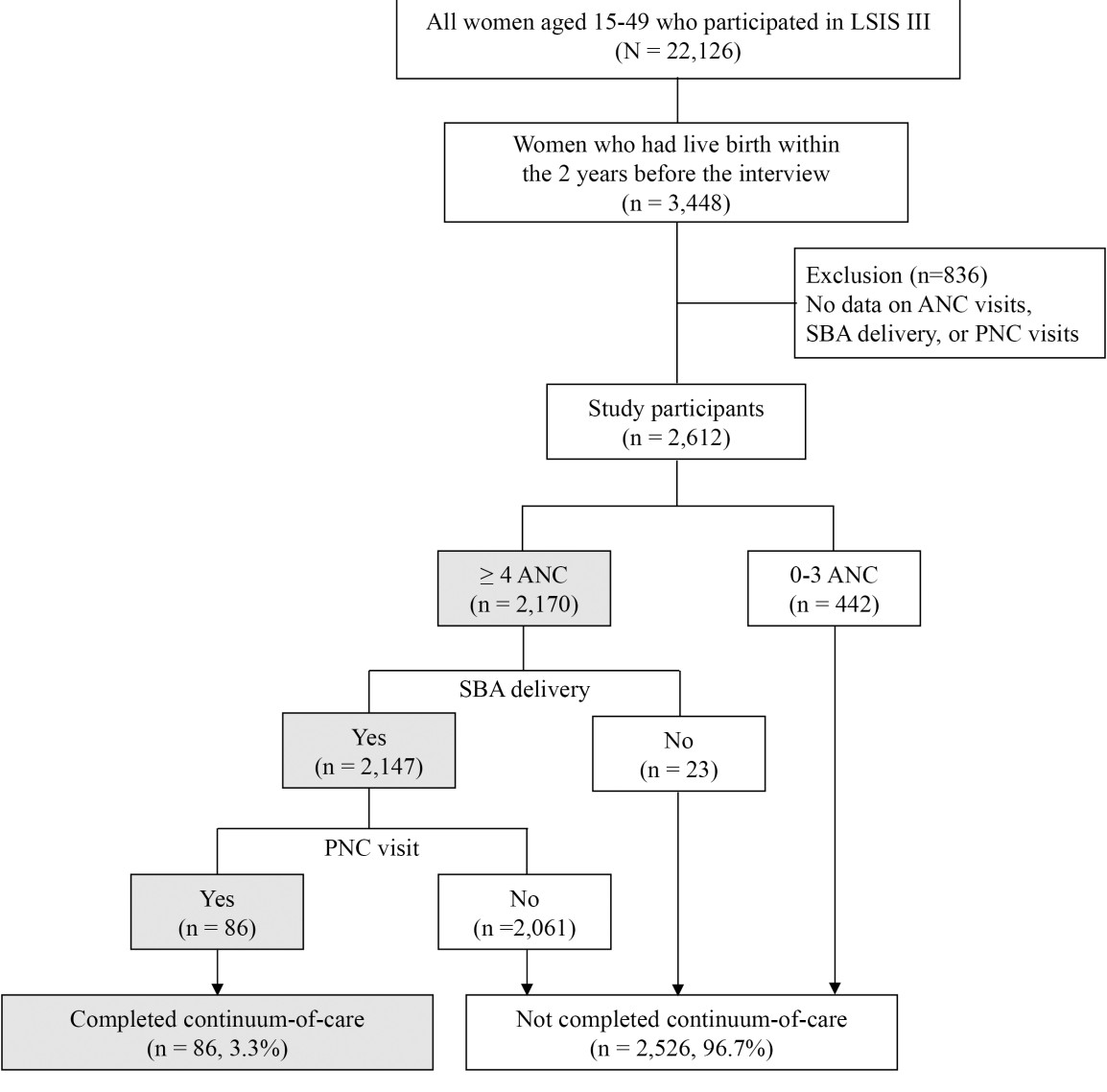

**Fig 1. Flowchart of the selection of study participants for the analysis of the maternal continuum-of-care.** Of the 22,126 women aged 15–49 years who participated in LSIS III (unweighted n = 22,126), 2,612 women were included in this study and 86 women completed the maternal continuum-of-care (weighted estimates). Except for the total number of LSIS III participants, all numbers shown in the figure are weighted to account for the complex survey design. LSIS, Lao Social Indicator Survey; ANC, antenatal care; SBA, skilled birth attendant; PNC, postnatal care.

married (divorced, widowed, or never married). "Never married" refers to women who reported neither current nor previous marital or cohabiting unions; this category may include women who experienced childbirth outside of marriage. Age at first marriage was categorized as "19 years or younger" and "20 years or older." Health insurance coverage was coded as "yes" if participants were covered by either the national health insurance scheme, the social health insurance scheme, or private insurance, and "no" if they had no coverage. The three schemes were combined because all provide financial protection for maternal and child health services, and analysis was not feasible due to the limited sample size within individual schemes.

## Obstetric characteristics

Obstetric variables included the number of children and characteristics related to the most recent pregnancy and delivery, such as number of ANC visits, gestational age at the first ANC visit, ANC provider, mode of delivery, delivery assistant, place of delivery, size of the last child at birth, sex of the last child, maternal health check before discharge from a health facility, and PNC visit after discharge within six weeks. The definitions and categorizations of these variables are summarized in S1 Table.

## Maternal continuum-of-care

In this study, the continuum-of-care was defined as receiving ANC at least four times, delivery attended by an SBA; including doctors, midwives, and nurses, and at least one PNC visit after discharge from a health facility within six weeks postpartum, as routine facility-based checks prior to discharge are nearly universal and do not capture continued care-seeking after delivery. Participants who received all three components were considered to have completed the continuum-of-care. Although the WHO recommends ≥ 8 ANC visits, this was not fully implemented in Lao PDR during the LSIS III period, and the national strategy still uses ≥ 4 ANC visits as an indicator. Due to the limited number of women who received PNC, having at least one PNC visit within six weeks after discharge was deemed sufficient to meet the PNC criterion. Although the continuum-of-care in this study did not distinguish between the types of providers, in Lao PDR, PNC is generally recommended to be provided by qualified healthcare providers, such as doctors, midwives, or nurses.

## Statistical analysis

Data were analyzed using the Statistical Package for the Social Sciences, version 27 (IBM Corp., Armonk, NY, USA). Sample weights were applied to generate population-representative estimates. Descriptive statistics were generated for all variables to provide an overview of the dataset. Chi-square or Fisher's exact tests were used to compare the characteristics of women who completed the continuum-of-care with those who did not. Univariate and multivariate logistic regression analyses were performed to assess associations between completion of the continuum-of-care and each variable. There were no missing data for any covariates included in the regression analyses. Odds ratios (ORs) and 95% confidence intervals (CIs) were calculated, with statistical significance set at $P < 0.05$.

Selection of variables for the multivariable model was determined based on prior literature and theoretical relevance, with the aim of adequately controlling for potential confounding factors. To assess potential multicollinearity among covariates, variance inflation factors (VIFs) were calculated for all variables included in the multivariable model. All VIF values were below commonly accepted thresholds, indicating no evidence of problematic multicollinearity. Model diagnostics were performed to evaluate the adequacy of the final model and model fit was assessed using the Hosmer–Lemeshow goodness-of-fit test.

## Ethical considerations

The LSIS III protocol was approved by the National Ethic Committee for Health Research in November 2022 [22]. Verbal informed consent was obtained from each participant. For participants aged 15–17 years, informed consent was obtained from their parent or guardian before obtaining their individual assent. Once consent was granted, the interviewers recorded it on the data collection form. This study is a secondary analysis of data from LSIS III, which are publicly available in anonymized form; therefore, additional ethical approval and participant consent for the current analysis were not required.

## Results

### Maternal care service status

A total of 2,612 women were included in the final analysis (Fig 1) after excluding 836 who did not respond to questions about the number of ANC and PNC visits. Among these women, 83.1% (n = 2,170) received at least four ANC visits, 98.6%

(n = 2,576) had delivery attended by an SBA, and 3.8% (n = 98) received at least one PNC visit after discharge. Overall, 82.2% (n = 2,147) received both ≥ 4 ANC visits and SBA-attended delivery. Only 3.3% (n = 86) completed the maternal continuum-of-care by receiving ≥ 4 ANC visits, SBA-attended delivery, and at least one PNC visit after discharge. The coverage declined markedly across the continuum, with the largest drop observed between SBA-attended delivery and PNC. Women who delivered at home (n = 736) were excluded from the analytical sample; therefore, all 2,612 women included in the analysis had facility-based deliveries. Characteristics of included and excluded women, including those with home deliveries, were compared and are presented in S2 Table.

## Socio-demographic characteristics of study participants

The largest age group was 20–29 years (55.5%, n = 1,447), followed by 30–39 years (28.5%, n = 745), and 15–19 years (13.7%, n = 362) (Table 1). Most women had received secondary education (47.8%, n = 1,250), were Lao-Tai (58.3%, n = 1,523), lived in rural areas (68.8%, n = 1,798), and were classified as belonging to the low/middle wealth index category (60.9%, n = 1,590). Of all the women, 97.2% (n = 2,538) were married, and 52.6% (n = 1,375) were married at age 19 years or younger. Health insurance coverage was 39.3% (n = 1,027).

## Obstetric characteristics of study participants

Most women had two or more children (58.3%, n = 1,523) (Table 2). Regarding ANC, 57.7% (n = 1,507) received 4–7 ANC visits, while 25.4% (n = 663) received 8 or more visits, and 63.5% (n = 1,659) had their first ANC visit at or before the 12th gestational week. Most women received ANC from a doctor (78.5%, n = 2,052), gave birth via vaginal delivery (90.4%, n = 2,360), had delivery attended by a doctor (80.4%, n = 2,100), and delivered at a hospital (70.8%, n = 1,850). Of the 3,448 women who had given birth within two years prior to the interview, 24.2% (n = 836) had delivered at home but were excluded from this study due to missing data on ANC and PNC visits. Most women gave birth to a child of average size (83.9%, n = 2,192), and 80.7% (n = 2,107) received a maternal health check before discharge from the health facility. However, only 3.8% (n = 98) had PNC visits after discharge within six weeks postpartum.

## Factors associated with completion of the maternal continuum-of-care

Binary logistic regression analysis (Table 3) showed that completion of the maternal continuum-of-care was associated with being aged 20–29 years (compared to 15–19 years), having post-secondary/tertiary education (versus none/elementary), classified as a high wealth index, being married at 20 years or older, having health insurance, receiving the first ANC visit at or before 12 weeks of gestation, having the last child as male, and receiving a maternal health check before discharge. In contrast, women from Mon-Khmer and Hmong-Mien ethnic groups (compared to Lao-Tai), and those who delivered at health centers (versus hospitals) were significantly less likely to complete the continuum-of-care.

Multivariate logistic regression analysis showed that being classified as having a high wealth index had approximately two times higher odds of completing the maternal continuum-of-care (adjusted odds ratio [AOR] = 2.19, 95% CI: 1.19–4.04, $P = 0.012$). Similarly, having the last child as male was associated with 1.65 times higher odds (AOR = 1.65, 95% CI: 1.04–2.61, $P = 0.032$), and receiving a maternal health check before discharge were associated with completion of the maternal continuum-of-care (AOR = 2.40, 95% CI: 1.06–5.41, $P = 0.035$). In contrast, Hmong-Mien women had statistically lower odds of completing the continuum-of-care than Lao-Tai women (AOR = 0.24, 95% CI: 0.06–0.91, $P = 0.036$). Sensitivity analyses were conducted to assess the robustness of the multivariable model. Alternative models, including only key covariates selected based on theoretical importance, were examined. The direction and magnitude of the main effect estimates were consistent with those observed in the primary model, indicating that the results were not materially affected by the number of covariates included.

**Table 1. Socio-demographic characteristics of study participants according to completion of the maternal continuum-of-care (n = 2,612).**

| Variable | Total | Continuum-of-care | | P-value |
|---|---|---|---|---|
| | | Completed | Not completed | |
| | (n = 2,612) | (n = 86) | (n = 2,526) | |
| | n (%) | n (%) | n (%) | |
| **Age (years old)** | | | | |
| 15–19 | 362 (13.7) | 4 (4.7) | 358 (14.2) | **0.030** |
| 20–29 | 1,447 (55.5) | 58 (67.4) | 1,389 (55.0) | |
| 30–39 | 745 (28.5) | 21 (24.4) | 724 (28.6) | |
| 40–49 | 58 (2.3) | 3 (3.5) | 55 (2.2) | |
| **Educational level** | | | | |
| None/elementary | 869 (33.3) | 22 (25.6) | 847 (33.5) | **<0.001** |
| Secondary | 1,250 (47.8) | 33 (38.4) | 1,217 (48.2) | |
| Post-secondary/tertiary | 493 (18.9) | 31 (36.0) | 462 (18.3) | |
| **Ethnicity** | | | | |
| Lao-Tai | 1,523 (58.3) | 69 (80.2) | 1,454 (57.5) | **<0.001** |
| Mon-Khmer | 642 (24.6) | 15 (17.5) | 627 (24.8) | |
| Hmong-Mien | 355 (13.6) | 2 (2.3) | 353 (14.0) | |
| Chinese-Tibetan | 73 (2.8) | 0 (0.0) | 73 (2.9) | |
| Other | 19 (0.7) | 0 (0.0) | 19 (0.8) | |
| **Residential region** | | | | |
| Northern | 820 (31.4) | 21 (24.4) | 799 (31.6) | 0.136 |
| Central | 1,333 (51.0) | 53 (61.6) | 1,280 (50.7) | |
| Southern | 459 (17.6) | 12 (14.0) | 447 (17.7) | |
| **Living area** | | | | |
| Urban | 814 (31.2) | 33 (38.4) | 781 (30.9) | 0.141 |
| Rural | 1,798 (68.8) | 53 (61.6) | 1,745 (69.1) | |
| **Wealth index quintile** | | | | |
| Low/middle | 1,590 (60.9) | 30 (34.9) | 1,560 (61.8) | **<0.001** |
| High | 1,022 (39.1) | 56 (65.1) | 966 (38.2) | |
| **Marital status** | | | | |
| Married | 2,538 (97.2) | 86 (100) | 2,452 (97.1) | 0.107 |
| Divorced/widowed/never married | 74 (2.8) | 0 (0.0) | 74 (2.9) | |
| **Age at first marriage (years old)** | | | | |
| ≤ 19 | 1,375 (52.6) | 31 (36.0) | 1,344 (53.2) | **0.002** |
| 20–49 | 1,237 (47.4) | 55 (64.0) | 1,182 (46.8) | |
| **Health insurance coverage** | | | | |
| No | 1,585 (60.7) | 43 (50.0) | 1,542 (61.1) | **0.039** |
| Yes | 1,027 (39.3) | 43 (50.0) | 984 (38.9) | |

## Discussion

This study examined the coverage and determinants of completion of the maternal continuum-of-care in Lao PDR using nationally representative LSIS III data. The findings revealed that the overall coverage of the maternal continuum-of-care was low (3.3%). Discontinuation of care most commonly occurred at the PNC stage. Factors associated with completion of the continuum-of-care included a higher wealth index, having the last child as male, and receiving a maternal health

**Table 2. Obstetric characteristics of study participants according to completion of the maternal continuum-of-care (n = 2,612).**

| Variable | Total | Continuum-of-care | | P-value |
|---|---|---|---|---|
| | | Completed | Not completed | |
| | (n = 2,612) | (n = 86) | (n = 2,526) | |
| | n (%) | n (%) | n (%) | |
| **Number of children** | | | | |
| 1 | 1,089 (41.7) | 43 (50.0) | 1,046 (41.4) | 0.136 |
| ≥ 2 | 1,523 (58.3) | 43 (50.0) | 1,480 (58.6) | |
| **Number of ANC visits** | | | | |
| 0–3 | 442 (16.9) | 0 (0.0) | 442 (17.5) | **<0.001** |
| 4–7 | 1,507 (57.7) | 43 (50.0) | 1,464 (58.0) | |
| 8 times or more | 663 (25.4) | 43 (50.0) | 620 (24.5) | |
| **Gestational age at the first ANC visit (weeks)** | | | | |
| ≤ 12 | 1,659 (63.5) | 67 (77.9) | 1,592 (63.0) | **0.005** |
| > 12 | 953 (36.5) | 19 (22.1) | 934 (37.0) | |
| **ANC provider** | | | | |
| Doctor | 2,052 (78.5) | 67 (77.9) | 1,985 (78.5) | 0.982 |
| Nurse/midwife | 527 (20.2) | 18 (20.9) | 509 (20.2) | |
| Other[a] | 33 (1.3) | 1 (1.2) | 32 (1.3) | |
| **Mode of delivery** | | | | |
| Vaginal delivery | 2,360 (90.4) | 68 (79.1) | 2,292 (90.7) | **<0.001** |
| Cesarean section | 252 (9.6) | 18 (20.9) | 234 (9.3) | |
| **Delivery assistant** | | | | |
| Doctor | 2,100 (80.4) | 67 (77.9) | 2,033 (80.5) | 0.361 |
| Nurse/midwife | 476 (18.2) | 19 (22.1) | 457 (18.1) | |
| Other[b] | 36 (1.4) | 0 (0.0) | 36 (1.4) | |
| **Place of delivery** | | | | |
| Hospital | 1,850 (70.8) | 70 (81.4) | 1,780 (70.5) | 0.079 |
| Health center | 743 (28.5) | 16 (18.6) | 727 (28.8) | |
| Private clinic | 19 (0.7) | 0 (0.0) | 19 (0.7) | |
| **Size of the last child at birth** | | | | |
| Large | 285 (10.9) | 10 (11.6) | 275 (10.9) | 0.715 |
| Average | 2,192 (83.9) | 70 (81.4) | 2,122 (84.0) | |
| Small | 135 (5.2) | 6 (7.0) | 129 (5.1) | |
| **Sex of the last child** | | | | |
| Male | 1,350 (51.7) | 54 (62.8) | 1,296 (51.3) | **0.036** |
| Female | 1,262 (48.3) | 32 (37.2) | 1,230 (48.7) | |
| **Maternal health check before discharge from the health facility** | | | | |
| Yes | 2,107 (80.7) | 79 (91.8) | 2,028 (80.3) | **0.008** |
| No | 505 (19.3) | 7 (8.1) | 498 (19.7) | |
| **PNC visits after discharge within six weeks postpartum** | | | | |
| Yes | 98 (3.8) | 86 (100.0) | 12 (0.5) | **<0.001** |
| No | 2,514 (96.2) | 0 (0.0) | 2,514 (99.5) | |

ANC, antenatal care; PNC, postnatal care.

[a]Other includes traditional birth attendants, village health volunteers, and others.

[b]Other includes traditional birth attendants, village health volunteers, relatives/friends, and others.

**Table 3. Bivariate and multiple logistic regression analysis on factors associated with completion of the maternal continuum-of-care among the participants (n = 2,612).**

| Variables | Unadjusted | | Adjusted | |
|---|---|---|---|---|
| | OR (95% CI) | *P*-value | AOR (95% CI) | *P*-value |
| **Age (years old)** | | | | |
| 15–19 | 1 (Reference) | | 1 (Reference) | |
| 20–29 | 3.51 (1.30-9.44) | **0.013** | 2.19 (0.74-6.48) | 0.157 |
| 30–39 | 2.44 (0.85-6.95) | 0.096 | 1.17 (0.34-4.09) | 0.806 |
| 40–49 | 4.38 (0.95-20.12) | 0.058 | 3.00 (0.55-16.41) | 0.206 |
| **Educational level** | | | | |
| None/elementary | 1 (Reference) | | 1 (Reference) | |
| Secondary | 1.09 (0.63-1.88) | 0.762 | 0.76 (0.42-1.39) | 0.371 |
| Post-secondary/tertiary | 2.65 (1.51-4.64) | **<0.001** | 1.25 (0.60-2.58) | 0.550 |
| **Ethnicity** | | | | |
| Lao-Tai | 1 (Reference) | | 1 (Reference) | |
| Mon-Khmer | 0.50 (0.28-0.88) | **0.016** | 0.81 (0.41-1.58) | 0.529 |
| Hmong-Mien | 0.15 (0.04-0.53) | **0.003** | 0.24 (0.06-0.91) | **0.036** |
| Chinese-Tibetan | – | – | – | – |
| **Residential region** | | | | |
| Northern | 1 (Reference) | | 1 (Reference) | |
| Central | 1.59 (0.95-2.66) | 0.078 | 1.31 (0.74-2.31) | 0.351 |
| Southern | 1.07 (0.53-2.19) | 0.845 | 0.74 (0.35-1.55) | 0.426 |
| **Living area** | | | | |
| Urban | 1 (Reference) | | 1 (Reference) | |
| Rural | 0.72 (0.46-1.11) | 0.138 | 1.42 (0.84-2.41) | 0.189 |
| **Wealth index quintile** | | | | |
| Low/middle | 1 (Reference) | | 1 (Reference) | |
| High | 3.07 (1.96-4.82) | **<0.001** | 2.19 (1.19-4.04) | **0.012** |
| **Age at first marriage (years old)** | | | | |
| ≤ 19 | 1 (Reference) | | 1 (Reference) | |
| 20–49 | 2.04 (1.31-3.20) | **0.002** | 1.08 (0.62-1.90) | 0.784 |
| **Health insurance coverage** | | | | |
| No | 1 (Reference) | | 1 (Reference) | |
| Yes | 1.58 (1.03-2.43) | **0.037** | 1.33 (0.84-2.13) | 0.228 |
| **Number of children** | | | | |
| 1 | 1 (Reference) | | 1 (Reference) | |
| ≥ 2 | 0.73 (0.47-1.12) | 0.143 | 0.86 (0.51-1.46) | 0.582 |
| **Gestational age at the first ANC visit (weeks)** | | | | |
| ≤ 12 | 2.06 (1.23-3.44) | **0.006** | 1.74 (0.99-3.05) | 0.054 |
| > 12 | 1 (Reference) | | 1 (Reference) | |
| **ANC provider** | | | | |
| Doctor | 1 (Reference) | | 1 (Reference) | |
| Nurse/midwife | 1.04 (0.61-1.77) | 0.881 | 1.58 (0.89-2.82) | 0.121 |
| Other[a] | 0.85 (0.11-6.92) | 0.882 | 2.87 (0.29-28.48) | 0.367 |
| **Place of delivery** | | | | |
| Hospital | 1 (Reference) | | 1 (Reference) | |
| Health center | 0.57 (0.33-0.99) | **0.045** | 1.00 (0.52-1.92) | 0.992 |
| Private clinic | – | – | – | – |

*(Continued)*

**Table 3.** (Continued)

| Variables | Unadjusted | | Adjusted | |
|---|---|---|---|---|
| | OR (95% CI) | *P*-value | AOR (95% CI) | *P*-value |
| **Size of the last child at birth** | | | | |
| Large | 1 (Reference) | | 1 (Reference) | |
| Average | 0.90 (0.46-1.77) | 0.765 | 0.92 (0.46-1.85) | 0.810 |
| Small | 1.36 (0.49-3.74) | 0.552 | 1.50 (0.52-4.32) | 0.454 |
| **Sex of the last child** | | | | |
| Male | 1.62 (1.04-2.53) | **0.034** | 1.65 (1.04-2.61) | **0.032** |
| Female | 1 (Reference) | | 1 (Reference) | |
| **Maternal health check before discharge from the health facility** | | | | |
| No | 1 (Reference) | | 1 (Reference) | |
| Yes | 2.61 (1.22-5.57) | **0.013** | 2.40 (1.06-5.41) | **0.035** |

OR, odds ratio; CI, confidence interval; AOR, adjusted odds ratio; ANC, antenatal care; PNC, postnatal care.

Hosmer–Lemeshow test was performed for the model (*P* = 0.216).

ᵃOther includes traditional birth attendants, village health workers, and others.

check before discharge. In contrast, Hmong-Mien women were less likely to complete the continuum-of-care than Lao-Tai women.

The coverage of the maternal continuum-of-care in this study was slightly lower than the coverage previously reported in Khammouane Province in 2016 (6.8%) [33], and substantially lower than reported in other LMICs in Southeast Asia, such as Indonesia and Cambodia [12,14]. Although definitions of continuum-of-care differed across studies, discontinuation of MNCH care most commonly occurred at the PNC stage. The low coverage of PNC may significantly contribute to the persistently high MMR observed in many LMICs [34], because PNC is a critical intervention for detecting maternal health complications following childbirth [3]. In Lao PDR, the significant lower utilization of PNC may be attributed to traditional postpartum practice of staying in a hot bed at home, the requirement of family permission to leave the house, or not receiving adequate information about PNC during ANC visits or at the time of delivery [35–37]. To improve PNC coverage, family engagement in maternal health should be promoted, accurate information regarding the importance of PNC should be provided, and implementation of standard discharge protocols, including the scheduling of PNC visits, should be strengthened [3,7,35,38].

Gender-related disparities in health service utilization have been reported in settings with son preference [39]. In such contexts, sons are often valued for family continuity and future economic support, which may influence family decision-making regarding pregnancy and childbirth care in Lao PDR [39,40]. Biological factors may also contribute, as male fetuses have reported to have higher risks of adverse birth outcomes, which may increase perceived pregnancy risk and influence care-seeking behaviors [32,41–43]. However, causal interpretations cannot be made given the cross-sectional design of this study.

An association between higher wealth status and the completion of the continuum-of-care has been reported in LMICs [19,20]. Despite the availability of free MNCH services for insured women, indirect costs such as transportation expenses and lost income remain important barriers to accessing healthcare in Lao PDR [35,44]. Uninsured women may face an even greater burden due to out-of-pocket payments. Concrete measures may include strengthening enrollment and continuity in the national health insurance scheme, providing transportation vouchers or conditional cash transfers for pregnant and postpartum women in rural or remote areas, and supporting community-based referral and transport mechanisms to facilitate timely access to health facilities. These approaches could help mitigate indirect costs and improve equitable access to MNCH services.

Women who received postpartum checks before discharge from health facilities may have been provided with comprehensive information, including PNC schedules [45]. Such information and counseling may facilitate subsequent utilization of PNC services, particularly when provided in local languages that women and their families can easily understand. In this study, 19.3% of women did not receive postpartum checks before discharge, although this is recommended in the national guidelines [31]. This may be related to early discharge, as LSIS III reported that 16.7% of women stayed at health facilities for less than 24 hours after delivery [22], although the WHO recommends a minimum stay of 24 hours [7]. Ensuring postpartum checks before discharge and promoting adequate post-delivery facility stays may help improve PNC utilization.

In this study, Hmong-Mien women were significantly less likely to complete the maternal continuum-of-care than Lao-Tai women. This disparity may be attributed to the fact that Hmong-Mien women are more likely to live in rural areas, have lower socioeconomic status, and face reduced access to healthcare services due to language barriers and dependence on family members for communication and decision making [23,46,47]. Moreover, Hmong-Mien women often prefer home births due to privacy concerns and cultural familiarity, tend to avoid male healthcare providers, and typically remain at home for one month postpartum as part of traditional practices [48,49]. These findings highlight the importance of culturally sensitive care and improved accessibility of maternal health services. Strategies such as communication in local languages, improved access to health facilities, and community-based outreach in collaboration with ethnic community leaders may be effective in promoting ANC and PNC utilization among Hmong-Mien women.

To improve coverage of the maternal continuum-of-care, village health volunteers may play a critical role in Lao PDR. Given the short duration of postpartum stays at health facilities and the low uptake of facility-based PNC, strengthening community-based follow-up is essential. Village health volunteers could support PNC utilization by providing health education to women and their families, identifying postpartum women in the community, and encouraging timely attendance at health facilities [50]. Close collaboration between healthcare providers and village health volunteers, including clear communication of PNC schedules at discharge, may help reduce drop-offs along the continuum-of-care, particularly in rural and remote areas [51].

This study has some strengths. First, this study used nationally representative LSIS III data, allowing assessment of the maternal continuum-of-care at the national level in Lao PDR. Second, the study examined multiple socio-demographic and obstetric factors associated with completion of the continuum-of-care. Third, the analysis provides important evidence to inform maternal health policy in Lao PDR, where empirical evidence on the continuum-of-care remains limited.

This study has some limitations. First, 836 of the 3,448 women who had given live birth within two years before the survey were excluded because of missing data on key variables. The excluded women were more likely to have characteristics associated with lower maternal healthcare utilization including home delivery, which may have resulted in an overestimation of the completion of the continuum-of-care. If all excluded women were classified as not completing the continuum-of-care, the overall completion rate would be 2.5% (86/3,448). The findings primarily reflect maternal healthcare utilization among facility-based births and may not be fully generalizable to women who had home delivery. Second, the definition of maternal continuum-of-care in this study is less stringent than current WHO recommendations regarding the number of ANC visits and was necessitated by data constraints and very low PNC uptake, which may limit comparability with other studies. The completion rate of the continuum-of-care when using at least ANC 8 visits was 1.7% (45/2,612). Third, the definition of maternal continuum-of-care in this study did not include maternal checks before discharge, because routine facility-based checks before discharge were considered nearly universal. However, 19.3% of the mothers in this study answered that they did not receive a maternal health check. Furthermore, the LSIS III data does not allow differentiation of PNC by location or precise timing, which limits the ability to fully characterize the continuum-of-care. Fourth, three health insurance schemes included in this study differ in their coverage and target populations; however, scheme-specific associations could not be examined because of the small number of women completing the continuum-of-care. Finally, causal relationships cannot be found in this study, as it is a cross-sectional study.

## Conclusion

The coverage of the maternal continuum-of-care among women with facility-based deliveries in Lao PDR was extremely low, only 3.3% in 2023, with a substantial gap between PNC utilization and the coverage of SBA delivery and ANC. Completion of the continuum-of-care was associated with higher wealth status, having a male child, and receiving a maternal health check before discharge. These findings highlight the need to strengthen maternal health checks before discharge, address gender norms and financial barriers, and promote culturally sensitive, community-based, and family-engaged approaches to improve the continuity of maternal healthcare in Lao PDR.

## Supporting information

**S1 Table. Definition and categorization of obstetric variables.**
(DOCX)

**S2 Table. Comparison of characteristics between included and excluded women.**
(DOCX)

## Acknowledgements

This study used secondary data from the Lao Social Indicator Survey III, 2023, which was implemented by the Lao Statistics Bureau and the Ministry of Health, with technical support from UNICEF and UNFPA. The authors extend their sincere gratitude to these organizations for making the data publicly available for research purposes.

## Author contributions

**Conceptualization:** Sengdavy Xaypadith, Eiko Yamamoto.

**Data curation:** Sengdavy Xaypadith.

**Formal analysis:** Sengdavy Xaypadith, Ai Aoki, Kimihiro Nishino, Eiko Yamamoto.

**Methodology:** Sengdavy Xaypadith, Ai Aoki, Kimihiro Nishino, Eiko Yamamoto.

**Supervision:** Eiko Yamamoto.

**Writing – original draft:** Sengdavy Xaypadith.

**Writing – review & editing:** Sengdavy Xaypadith, Ai Aoki, Kimihiro Nishino, Eiko Yamamoto.

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
