## [Decision Letter · Decision Letter 0]

23 Dec 2025

Dear Dr. Yamamoto,

Thank you for submitting your manuscript to PLOS ONE. After careful consideration, we feel that it has merit but does not fully meet PLOS ONE’s publication criteria as it currently stands. Therefore, we invite you to submit a revised version of the manuscript that addresses the points raised during the review process.

We look forward to receiving your revised manuscript.

Kind regards,

Kanchan Thapa, MPH, MPhil

Academic Editor

PLOS One

Additional Editor Comments (if provided):

Reviewers' comments:

Reviewer's Responses to Questions

**Comments to the Author**

1. Is the manuscript technically sound, and do the data support the conclusions?

Reviewer #1: Partly

Reviewer #2: Yes

Reviewer #3: Yes

Reviewer #4: Yes

Reviewer #5: Yes

Reviewer #6: Yes

Reviewer #7: Yes

Reviewer #8: Yes

Reviewer #9: Partly

2. Has the statistical analysis been performed appropriately and rigorously?

Reviewer #1: No

Reviewer #2: Yes

Reviewer #3: Yes

Reviewer #4: Yes

Reviewer #5: Yes

Reviewer #6: Yes

Reviewer #7: Yes

Reviewer #8: Yes

Reviewer #9: No

3. Have the authors made all data underlying the findings in their manuscript fully available?

Reviewer #1: Yes

Reviewer #2: Yes

Reviewer #3: Yes

Reviewer #4: Yes

Reviewer #5: Yes

Reviewer #6: Yes

Reviewer #7: Yes

Reviewer #8: Yes

Reviewer #9: Yes

4. Is the manuscript presented in an intelligible fashion and written in standard English?

Reviewer #1: Yes

Reviewer #2: Yes

Reviewer #3: Yes

Reviewer #4: Yes

Reviewer #5: Yes

Reviewer #6: Yes

Reviewer #7: Yes

Reviewer #8: Yes

Reviewer #9: Yes

Reviewer #1: The study is on cross-sectional study that focuses on maternal continuum-of-care coverage in Lao PDR based on nationally representative data. Extremely low completion rates (3.3) and related factors are the result of the authors. Although the subject is significant, and the data are strong, several methodological and interrogatory issues should be resolved.

1. It is very large (30.7% of the participants) that 1,156 women were not included because of missing data, and this is not sufficiently discussed. The authors admit that these were mostly home deliveries; however, they do not examine the nature of the disqualified women and leave it unnoticed how this impacts the results. This highly restricts the ability to generalize, because the research, in essence, only analyzes facility-based births and concludes on coverage in the nation. Perform sensitivity analysis of the features of included and excluded women. Make it clear in the abstract and other conclusions that findings are relevant to facility-based births.

2. The definition applied (≥4 ANC visits, SBA delivery, ≥1 PNC visit) is not as rigorous as the WHO recommendations (8 ANC visits, PNC at specific intervals). Although the authors explain this by the low PNC uptake, a low threshold can inflate actual continuum completion and restrict the comparison with other research. Additional analysis with WHO-suggested thresholds or limitations should be mentioned, or limitations should be discussed more clearly.

3. There is a total of 11 variables and only 86 outcomes in the multivariate model, which can overfit (ratio of 8 events per variable). There is no multicollinearity assessment discussion. Not describe the variable selection process. Reduce adjusted selection of variables, report model diagnostics, and think about a more parsimonious model.

4. The terms associated with and factors, throughout the manuscript, imply causation despite the cross-sectional design. Indicatively, such as the fact that having the last child as a male was linked to completion indicates that male child sex implies care-seeking, yet one cannot tell whether it was the timing. Be more careful in language, and focus on correlational results.

5. PNC variable of interest includes "at least one PNC visit in the six weeks following discharge," but does not differentiate between facility-based PNC and home-based PNC, and the timing of the visit(s). This curtails the meaning of the "continuum."

6. The theme of a gender preference (lines 284-285) should be discussed more subtly. The male-child care completion relationship may represent various processes other than the cultural preference for sons.

7. No data were given on the handling of missing data on covariates in regression models. Did it exclude cases that had any covariate data that were missing?

8. The variable of health insurance is an unreasonable combination of two schemes. These can include other levels of coverage and, therefore, may influence the use of care differently.

• Table 3: It might be worthwhile to drop non-significant variables from the adjusted model or stepwise.

• The SBA coverage of 98.6 percent does not appear to be in line with 827 home deliveries among excluded women.

• 222: Expand upon, all 2612 women delivered at the health facility, since home deliveries have already been mentioned above.

9. The discussion may be made shorter. Certain paragraphs represent result information (e.g., lines 258-266).

• Missing affiliation number 4 - The sequence of check numbering.

• 269: postpartum hemorrhage is the leading cause - must be cited.

• Surely, Tables 1 and 2 are worth combining or shifting to the annexes.

• Abstract: Indicate that this analysis is limited to the facility-based births.

Reviewer #2: This manuscript addresses an important public health issue: the very low completion of the maternal continuum-of-care (CoC) in Lao PDR. The paper is clearly structured, uses nationally representative data (LSIS III), and applies appropriate statistical methods. The major strengths are the use of high-quality population survey data and the identification of socio-cultural determinants—including ethnicity, wealth, and gender norms.

Overall, the study is valuable and relevant, but revisions would improve its rigor and clarity.

1. Sample selection and potential bias

A large proportion of eligible women (approximately 31%) were excluded due to missing ANC/PNC data, primarily among those who delivered at home. This exclusion likely introduces systematic bias and may underestimate true CoC coverage. Can you describe the characteristics of excluded vs. included women (if available), the reason for missingness in LSIS III and how this exclusion may influence the results and their generalizability?

2. Definition of continuum-of-care

The study uses ≥4 ANC visits as the ANC criterion, whereas current WHO guidance recommends ≥8 contacts. Please provide a justification why you select the 4-visit threshold.

3. Multivariate Model Specification Needs More Explanation

The selection of variables for the multivariate model is not fully transparent. Please clarify which criteria us used (e.g., all variables with P<0.2 in univariate analysis were included), or whether theoretical justification was used. It is important to consider reporting model diagnostics (multicollinearity, goodness-of-fit). In results section, it is good to use " how many times higher or lower" instead of jargon (aOR).

Minor Issues

1. Correct typographical errors in tables (e.g., % of Non/ Elementary Education Level should be 33.3%. in Table 1).

2. Ensure Figure 1 is clearly labeled, legible, and matches the text description.

3. Table 2 footnote: “Other includes traditional birth attendants and village health volunteers”—is needed to be consistent with explanation in Line no. 150.

4. Standardize terminology (e.g., “upper/high wealth index” in Line no. 338).

Reviewer #3: Reviewer Comments on “The coverage of maternal continuum-of-care and associated factors in the Lao People’s Democratic Republic: A population-based cross-sectional study.”

Thank you for the opportunity to review this manuscript. The study found that maternal continuum-of-care coverage in Lao PDR remains very low, with significant gaps across antenatal, delivery, and most remarkably postnatal services, and is influenced by socioeconomic, demographic, and geographic factors.

Using secondary data from the Lao Social Indicator Survey III (LSIS III, 2023), researchers conducted a population-based cross-sectional analysis to assess the extent of maternal continuum-of-care (CoC) - defined as receiving antenatal care (ANC), skilled birth attendance (SBA), and postnatal care (PNC). The findings revealed that only a small proportion of women completed the full continuum, with drop-offs most pronounced between SBA and PNC.

Abstract: “Of the 2,612 women, 83.1% had at least four ANC visits, 98.6% had SBA delivery, 3.8% had at least one PNC visit, and 3.3% completed the maternal continuum-of-care.” There may need to be added conjunctions like “of which 98.6% had SBA delivery, of which 3.8% had at least one PNC visit, …” or something like this, to make visible that these proportions were continuous or cascaded.

Study Design: Line 105 - Sample size (15-49 women) in LSIS III was 22,126. Is there any explanation why some were excluded from 22,512? It would be better to describe it shortly here.

Line 112 - The proportion of deliveries within two years looks quite small at about 17%. Any comparison within the country or regions in recent years stated in LSIS III? It would be better to describe the other RH indicators.

Line 112 - In this study, “Women who had childbirth within the 2 years before the interview (N= 3,768)”, I have found that the Number of women with a live birth in the last 2 years was 3448 in the Tables of ANC and PNC. Stillbirths included in your study with 3768 women? If so, could you please explain more? Usually, CoC completion is with live births, but in some cases, SB were included with explanation.

Among 3768, 1156 were excluded due to data incompleteness; about 30.7% of the women who had given birth within 2 years were excluded. It is doubtful about the data quality in LSIS III. To prove the data quality, do you have any more justification for this, apart from the home deliveries that you have explained in the limitations?

Results: Is there any data disaggregated by duration since child birth – within 6 months, 6-12 months, 12-18 months, and 18-24 months? It is interesting to know the differences within these groups.

CoC coverage – from the service delivery point of view, level or rank of health staff – which may have different coverage or proportion in drop-offs within healthcare providers, so that we can know which provider should be built capacity in priorities. As the coverage is very low, we need to explore the service provider’s site as well.

Authors may provide the proportion of drop-offs across the CoC: ANC – SBA delivery – PNC, and compare with those in LSIS if possible.

Tables 1 and 2 – why authors put column percentages? The variable of interest is whether CoC is complete or not; we can compare across the categories of independent variables by row percentages, which is more informative. Or, why do you choose column percentage? Please give justification. The table data explanation also emphasizes only frequency within variables, no description related to CoC Yes or No, with the p-value (Chi-Square Test results in tables), whether significant or not. It is better to describe them here.

Conclusion: Do you have any recommendations for healthcare providers’ sites?

Reviewer #4: The manuscript addresses an important public health issue by examining the coverage of the maternal continuum of care (CoC) in the Lao People’s Democratic Republic (Lao PDR). The use of a population-based, cross-sectional study design is appropriate for estimating coverage proportion and identifying determinant factors. It could contribute valuable insights for informing national maternal health strategies and policy prioritization. Here are specific recommendations for the authors to consider:

1. In the introduction section, revise the 1st and 2nd paragraphs to be a logical flow of the problem statement. For example, Line No. 58-59, sudden transition to the SDG targets without joint information makes it abrupt and confused for the readers. Structural improvement is needed in these phrases.

2. In the methods section, while describing the socio-demographic factors (Pg No. 6, Line no. 131-132), it’s better to recheck the wealth index quintiles (low, middle, upper, and high). The first thing is that if the authors express wealth index in quintiles, it should be 5 groups. If only four groups, it should be quantile. Please make it clear since it could change the final results. Also, check the original dataset and confirm how they mention the wealth status in their report.

3. In the methods section, while describing the obstetric factors (Pg No. 7, Line no. 151-155), delivery assistant includes “other” category referring to traditional birth attendants and village health volunteers. However, in the place of delivery, the categories are “hospital,” “health center,” or “private clinic” only. It makes a little bit confused whether home delivery is included or not. Please recheck the data and add this group if relevant.

4. In the methods section, while describing the maternal CoC (Pg No. 8, Line no. 162), the CoC was defined as receiving ANC at least four times. Meanwhile. the WHO's current recommendation is eight ANC contacts. Here, the authors should clearly articulate the rationale for this specific operational definition in the context of Lao PDR's national guidelines, if they differ from the WHO standard. Moreover, in Line No. 166-167, it’s better to mention types of healthcare provider for standard postpartum care in the operational definition.

5. In the methods section, while statistical analysis (Pg No. 8), it would be beneficial to add a diagnostic check for multicollinearity among the predictor variables used in the final adjusted model. Specifically, please calculate and report the Variance Inflation Factor (VIF) for each variable in the multivariable logistic regression. This step is crucial for confirming that the standard errors and p-values are not disproportionately inflated due to linear dependencies among the covariates

6. In the results section, (Pg No. 9, Line no. 195-199), most information is repeatedly described, better to remove them and rephrase to become more concise results.

7. 6. Please re-test the statistical analysis and rearrange the tables if the authors need to revise the variables “wealth index quintiles” and “place of delivery”.

8. The discussion and conclusion parts are well constructed with relevant contrast.

Reviewer #5: This manuscript addresses an important maternal health issue and provides valuable insights into antenatal care (ANC) and postnatal care (PNC) utilization among women in the Lao People’s Democratic Republic. The topic is highly relevant to public health programs, particularly in low- and middle-income countries where barriers to maternal health services remain significant. The study objectives are clearly stated, and the use of population data provides useful implications for service improvement. However, there are some areas that require additional clarification and refinement to enhance the accuracy, interpretability, and practical value of the findings.

Reviewer #6: This study is valuable and may stimulate government officials to improve maternal, newborn, and child health in developing countries worldwide. Factors influencing the low completion of the maternal continuum of care can be potentially applied to improve maternal, newborn, and child health in the study area.

Reviewer #7: 1. The study is well designed, and the data support the authors’ conclusions.

2. The authors used the right statistical methods and analyzed the data carefully.

3. The data used in the study are accessible to the public through the MICS website.

4. The paper is easy to understand and written in acceptable English.

5. The study is important for improving maternal health in Lao PDR. The authors mention the study’s limitations and give useful recommendations. They should explain how to address cultural barriers and how their findings can guide policy. Overall, the paper is clear and useful.

6. The manuscript identifies that Hmong-Mien women are significantly less likely to complete the continuum of care, mentioning cultural practices and language barriers. However, the discussion remains general and does not provide actionable recommendations or detailed strategies to address these barriers.

7. While the manuscript recommends strengthening health insurance and reducing financial barriers, it does not offer concrete examples of how these can be implemented or scaled up in Lao PDR.

8. The paper highlights low postnatal care (PNC) utilization and the importance of family engagement but does not cite or describe successful programs from other contexts that could serve as models.

9. The authors note that women who delivered at home were excluded due to missing data but do not discuss how this exclusion may bias the results or affect generalizability.

10. While the manuscript is generally clear, it will be better after minor revisions, such as lengthy explanations and a listing of variables in the methods section.

11. Since statistically significant results are not visually distinguished from non-significant ones in Tables 1, 2, and 3, it needs to consider bolding or marking statistically significant p-values or using symbols (e.g., * for p < 0.05).

12. It should expand the caption to clarify what Table 3 shows (e.g., “Table 3. Bivariate and multiple logistic regression analysis of factors associated with completion of the maternal continuum-of-care among the participants (N=2,612)”).

Reviewer #8: How exactly was PNC defined in LSIS III and in your analysis? Does LSIS capture immediate postnatal checks before discharge? If yes, please include it or justify exclusion.

Did you account for survey strata and cluster/PSU in SPSS (Complex Samples)? Please specify the variables and procedures used.

Please provide a weighted comparison of included vs excluded women (n=2,612 vs n=1,156 excluded) to assess selection bias.

With only 86 events, please justify the multivariable model complexity and report events-per-variable; consider a parsimonious model or penalized methods.

Please correct Table 1 education percentage (869 cannot be 3.33% of 2,612) and re-check all tables.

Please provide the ethics approval number for LSIS III (or explain why unavailable).

Reviewer #9: Regarding the outcome measures, line number 162-167 say “the continuum-of-care was defined as receiving ANC at least four times, delivery attended by an SBA; including doctors, midwives, and nurses, and at least one PNC visit after discharge from a health facility within six weeks postpartum. Participants who received all three components were considered to have completed the continuum-of-care. Due to the limited number of women who received PNC, having at least one PNC visit within six weeks after delivery was deemed sufficient to meet the PNC criterion.”

So, among 2617 study participants, 2170 got more than 4 ANC visits, 2147 got SBA delivery but only 86 got PNC. Given that nearly all women delivered in health facilities (n = 2,612), including 252 cesarean sections, it is unclear whether women who had SBA-assisted institutional deliveries did not receive any postnatal assessment, including within 24 hours of delivery. Postnatal care was operationalized as a separate maternal health check occurring after discharge from the delivery facility. This strongly suggests that routine postpartum checks before discharge were not classified as PNC.

This results low prevalence of complete continuum of care (3.3%) largely due to the restricted operational definition of postnatal care rather than a true absence of services, which may affect the outcome estimates and may create a highly imbalanced outcome distribution which could affect the precision and stability of multivariable regression estimates.

As a result, the low outcome prevalence (3.3%) leads to a highly imbalanced outcome distribution and a relatively small number of events despite the large study population. Under these conditions, conventional logistic regression would be prone to small-sample bias, overfitting of predictors, and an increased risk of type I error. Reducing the number of predictors or applying penalized logistic regression methods is therefore recommended to minimize overfitting and prevent inflation of odds ratio estimates.

**Do you want your identity to be public for this peer review?** For information about this choice, including consent withdrawal, please see our Privacy Policy

Reviewer #1: **Yes:** Dr. Jonah Bawa Adokwe

Reviewer #2: No

Reviewer #3: No

Reviewer #4: No

Reviewer #5: **Yes:** Kyaw Zayar Aung

Reviewer #6: No

Reviewer #7: **Yes:** Tun Win Lat

Reviewer #8: No

Reviewer #9: No

---

## [Author Response · Author response to Decision Letter 1]

4 Feb 2026

Response to the editor’s and reviewers’ comments

We would like to thank the editor and reviewers for reviewing our manuscript. We have revised the manuscript according to your comments, which were very helpful. The revised manuscript has been proofread by a native English speaker. The revisions have been completed and our responses are as follows.

Author response:

We have carefully revised the manuscript to ensure that it meets PLOS ONE’s style requirements.

Author response:

Thank you for your guidance regarding the Data Availability requirements.

In response to your comment, we have uploaded the minimal data set necessary to replicate the findings of our study to a stable, public repository in accordance with PLOS ONE’s data sharing standards. Specifically, the derived and fully anonymized dataset used for the analyses has been deposited in figshare.

Repository: figshare

DOI: https://doi.org/10.6084/m9.figshare.31145893

This dataset represents the minimal data set required to reproduce the results reported in the manuscript and does not contain any personally identifiable information. The data were derived from the Lao Social Indicator Survey III (LSIS III), conducted as part of UNICEF’s Multiple Indicator Cluster Surveys (MICS).

Lines 217-218: We have revised the Data Availability statement in the Methods section of the manuscript to reflect this update. The revised statement now indicates that the minimal dataset is publicly available via figshare, while the original LSIS III data remain accessible from the UNICEF MICS repository upon user registration and approval, in accordance with their data use policy. We hope that these revisions fully address the journal’s requirements.

Author response:

We have carefully checked the manuscript and confirmed that the ethics statement appears only once, under the “Ethical considerations” subsection within the Methods section. There is no duplicate ethics statement in any other section of the manuscript.

Author response:

There was no comment from the reviewers that includes a recommendation to cite specific previously published works.

Reviewer #1: The study is on cross-sectional study that focuses on maternal continuum-of-care coverage in Lao PDR based on nationally representative data. Extremely low completion rates (3.3) and related factors are the result of the authors. Although the subject is significant, and the data are strong, several methodological and interrogatory issues should be resolved.

1-1. It is very large (30.7% of the participants) that 1,156 women were not included because of missing data, and this is not sufficiently discussed. The authors admit that these were mostly home deliveries; however, they do not examine the nature of the disqualified women and leave it unnoticed how this impacts the results. This highly restricts the ability to generalize, because the research, in essence, only analyzes facility-based births and concludes on coverage in the nation. Perform sensitivity analysis of the features of included and excluded women. Make it clear in the abstract and other conclusions that findings are relevant to facility-based births.

Author response:

Thank you for this important and constructive comment. We applied sampling weights for all analyses and Fig 1 has been revised. Of 3,448 women (weighted) had given birth in the past 2 years, 2,612 (weighted) were included and 836 (weighted) were excluded.

We conducted a sensitivity analysis comparing women included in the final analysis with those excluded due to missing data on key outcome variables. The results of this analysis are presented in a new table (Suppl Table). Statistically significant differences were observed between the two groups across most socio-demographic and obstetric characteristics. Excluded women were more likely to have no or an elementary education, belong to ethnic minority groups as Mon-Khmer, reside in rural areas, be classified as low/middle wealth index, be married at age 19 years or younger, and have no health insurance. Nearly all excluded women delivered at home.

Lines 387-398: In response, we have revised the Limitations section to clearly state that the findings primarily reflect patterns of maternal healthcare utilization among women who delivered in health facilities and may not be generalizable to home deliveries as follows, “This study has some limitations. First, 836 of the 3,448 women who had given live birth within two years before the survey were excluded because of missing data on key variables. The excluded women differed systematically from those included, being predominantly home births and more likely to have low socioeconomic status, including having lower educational attainment, belonging to ethnic minority groups, residing in rural areas, lower/middle wealth index, being married at age 19 years or younger, and having no health insurance (Suppl Table). As these characteristics are associated with lower utilization of maternal healthcare services, this exclusion may have resulted in an overestimation of the completion of the continuum-of-care. Consequently, the findings primarily reflect maternal healthcare utilization among facility-based births and may not be fully generalizable to women who had home delivery. For contextual reference, if all excluded women were classified as not completing the continuum-of-care, the overall completion rate would be 2.5% (86/3,448).”

1-2. The definition applied (≥4 ANC visits, SBA delivery, ≥1 PNC visit) is not as rigorous as the WHO recommendations (8 ANC visits, PNC at specific intervals). Although the authors explain this by the low PNC uptake, a low threshold can inflate actual continuum completion and restrict the comparison with other research. Additional analysis with WHO-suggested thresholds or limitations should be mentioned, or limitations should be discussed more clearly.

Author response:

Lines 181-183: Thank you for the comment. We acknowledge that the definition of maternal continuum-of-care used in this study is less strict than current WHO recommendations, which include at least eight antenatal contacts and PNC at specific time intervals. Due to limitations in available LSIS III variables and the extremely low uptake of PNC, applying WHO-recommended thresholds was not feasible in this study. Regarding the number of ANC visits, we have added a sentence to the Methods section as follows, “Although the WHO recommends ≥ 8 ANC contacts, this was not fully implemented in Lao PDR during the LSIS III period, and the national strategy still uses ≥ 4 ANC visits as an indicator.”

Lines 398-402: We have revised the Limitations section to clearly state that the use of lower thresholds may overestimate continuum-of-care completion and restrict comparability with studies using WHO-recommended definitions as follows, “Second, the definition of maternal continuum-of-care in this study is less stringent than current WHO recommendations regarding the number of ANC visits and was necessitated by data constraints and very low PNC uptake, which may limit comparability with other studies. Completion rate of the continuum-of-care when using at least ANC 8 visits was 1.7% (45/2,612).”

1-3. There is a total of 11 variables and only 86 outcomes in the multivariate model, which can overfit (ratio of 8 events per variable). There is no multicollinearity assessment discussion. Not describe the variable selection process. Reduce adjusted selection of variables, report model diagnostics, and think about a more parsimonious model.

Author response:

We thank the reviewer for raising important concerns regarding potential overfitting and model specification.

We acknowledge that the multivariable model includes 11 variables with 86 outcome events, which may raise concerns about the events-per-variable ratio. However, all variables were selected a priori based on established literature and theoretical relevance to control for potential confounding, rather than through data-driven selection procedures.

To address concerns about model stability, we assessed multicollinearity among all covariates using VIFs. All VIF values (1.009-1.743) were below commonly accepted thresholds (< 5.0), indicating no evidence of problematic multicollinearity. In addition, we conducted sensitivity analyses using alternative, more parsimonious models including only key covariates. The direction and magnitude of the main effect estimates were consistent with those of the primary model, supporting the robustness of our findings.

Lines 199-205: We have added sentences in the Methods section as follows, “Selection of variables for the multivariable model was determined a priori based on prior literature and theoretical relevance, with the aim of adequately controlling for potential confounding factors. To assess potential multicollinearity among covariates, variance inflation factors (VIFs) were calculated for all variables included in the multivariable model. All VIF values were below commonly accepted thresholds, indicating no evidence of problematic multicollinearity. Model diagnostics were performed to evaluate the adequacy of the final model and model fit was assessed using the Hosmer–Lemeshow goodness-of-fit test.”

Lines 282-287: We have added sentences in the Results section as follows, “Sensitivity analyses were conducted to assess the robustness of the multivariable model. Alternative models, including only key covariates selected based on theoretical importance, were examined. The direction and magnitude of the main effect estimates were consistent with those observed in the primary model, indicating that the results were not materially affected by the number of covariates included.”

We believe these additional analyses and clarifications adequately address the reviewer’s concerns regarding overfitting and model robustness.

1-4. The terms associated with and factors, throughout the manuscript, imply causation despite the cross-sectional design. Indicatively, such as the fact that having the last child as a male was linked to completion indicates that male child sex implies care-seeking, yet one cannot tell whether it was the timing. Be more careful in language, and focus on correlational results.

Author response:

Thank you for this important comment. We agree that causal language should be avoided considering the cross-sectional design. We have revised the manuscript throughout to replace causal wording with correlational language.

1-5. PNC variable of interest includes "at least one PNC visit in the six weeks following discharge," but does not differentiate between facility-based PNC and home-based PNC, and the timing of the visit(s). This curtails the meaning of the "continuum."

Author response:

We thank the reviewer for this thoughtful comment regarding the definition of PNC and its implications for the concept of the continuum of care. In this study, PNC was defined as having at least one PNC visit within six weeks following discharge from a health facility. This definition was adopted because facility-based maternal health checks before discharge are routinely provided following facility delivery in Lao PDR and therefore do not meaningfully differentiate PNC utilization after childbirth. Our focus was thus on post-discharge care, which reflects continued engagement with maternal health services beyond delivery. We acknowledge that the available LSIS III data do not allow further differentiation between facility-based and home-based PNC, nor do they capture the precise timing of PNC visits within the six-week period. We agree that this limits the granularity with which the continuum-of-care can be characterized. We have clarified this rationale in the Methods section and explicitly acknowledged this limitation in the Discussion. Despite these constraints, we believe that the post-discharge PNC indicator remains a meaningful measure of continuity of maternal health care within the context of the available data. We have revised the manuscript to more clearly describe the scope and limitations of this definition.

Lines 176-180: We have revised the sentence in the Methods section as follows, “the continuum-of-care was defined as receiving ANC at least four times, delivery attended by an SBA; including doctors, midwives, and nurses, and at least one PNC visit after discharge from a health facility within six weeks postpartum, as routine facility-based checks prior to discharge are nearly universal and do not capture continued care-seeking after delivery.”

Lines 402-408: We have added a sentence in the Discussion section as follows, “Third, the definition of maternal continuum-of-care in this study included post-discharge PNC but not maternal check before discharge, because routine facility-based checks before discharge were considered nearly universal. However, 19.3% of the mothers in this study answered that they did not receive a maternal health check. Furthermore, the LSIS III data does not allow differentiation of PNC by location or precise timing, which limits the ability to fully characterize the continuum-of-care.”

1-6. The theme of a gender preference (lines 284-285) should be discussed more subtly. The male-child care completion relationship may represent various processes other than the cultural preference for sons.

Author response:

Lines 316-327: Thank you for this important comment. We have revised the Discussion section to address this concern by adopting a more nuanced interpretation of the observed association as follows, “Gender-related disparities in health service utilization have been reported in settings with son preference [40]. Sons are often viewed as important for family continuity and as a source of labor in rural settings, and, in contexts where formal social security systems are limited, they may be perceived as providing future support in old age [40]. In Lao PDR, cultural norms and gender roles may still influence family decision-making regarding pregnancy and childbirth care [41]. In addition, biological and clinical factors may also contribute to this association. Previous studies have reported a higher incidence of stillbirth, neonatal morbidity and mortality, preterm delivery, and congenital anomalies among male fetuses compared with females [32, 42-44], which may increase perceived pregnancy risk and influence care-seeking behaviors. Taken together, these findings suggest that the observed association likely reflects a combination of social, cultural, and biological factors, and causa

---

## [Editor Report · Decision Letter 1]

6 Mar 2026

Dear Dr. Yamamoto,

Thank you for submitting your manuscript to PLOS ONE. After careful consideration, we feel that it has merit but does not fully meet PLOS ONE’s publication criteria as it currently stands. Therefore, we invite you to submit a revised version of the manuscript that addresses the points raised during the review process.

plosone@plos.org . A letter that responds to each point raised by the academic editor and reviewer(s). You should upload this letter as a separate file labeled 'Response to Reviewers'.A marked-up copy of your manuscript that highlights changes made to the original version. You should upload this as a separate file labeled 'Revised Manuscript with Track Changes'.An unmarked version of your revised paper without tracked changes. You should upload this as a separate file labeled 'Manuscript'.

We look forward to receiving your revised manuscript.

Kind regards,

Kanchan Thapa, MPH, MPhil

Academic Editor

PLOS One

Journal Requirements:

Additional Editor Comments :

Line 107: Please change to Study participants

Line 123: Proof read the line

Line 129: I suggest to change to- Socio-demographic characteristics

Line 150 and following paragraph- Tighten your language, I may suggest to include all the variables into a single table or subheading including all the variable definition.

Line 223:

I suggest to change to – Maternal Care service status

Line 237- I suggest to change to - Socio-demographic factors of the study subjects into Socio-demographic Characteristics of the study participants

Table 1. use n= ....rather than N and format table as per PLOS One standards

Line 247: the title is not representing the table you are presenting. You are showing the Cross tabulation, please review similar paper and format accordingly. I shall suggest to take consultation with an experienced statistician.

Line 266: the sub-heading should be something like- Association between socio-demographic factors and completion of maternal continuum of care rather than - Logistic regression analysis on completion of the maternal 267 continuum-of-care

Table 3. is perfect but format for clarity and we dont need P value here. In the footnote highlight which aOR or cOR is significant based on OR

Discussion

First paragraph overall summarize your study findings. I suggest to rewrite your whole section analytically.

Your discussion is too long and loosely written, I suggest to rewrite analytically and tighten the results.

Where is your strength and limitation of the study, please include these too.

Conclusion

Rework on conclusion, your writing need to tighten. Conclude based on the information.

---

## [Author Response · Author response to Decision Letter 2]

8 Mar 2026

Response to the editor’s and reviewers’ comments

We would like to thank the editor and reviewers for reviewing our manuscript. We have revised the manuscript according to your comments, which were very helpful. The revised manuscript has been proofread by a native English speaker. The revisions have been completed and our responses are as follows.

Author response:

There was no comment from the reviewers that includes a recommendation to cite specific previously published works.

Additional Editor Comments:

1. Line 107: Please change to Study participants

Author response:

Lines 107: We have revised “Study subjects” to “Study participants.”

2. Line 123: Proof read the line

Author response:

Line 123: Thank you for the suggestion. We have revised to “Of the 22,126 women aged 15–49 years who participated in LSIS III (unweighted n = 22,126).”

3. Line 129: I suggest to change to- Socio-demographic characteristics

Author response:

Line 129: Thank you for the suggestion. We have revised “Socio-demographic factors” to “Socio-demographic characteristics.”

4. Line 150 and following paragraph- Tighten your language, I may suggest to include all the variables into a single table or subheading including all the variable definition.

Author response:

Lines 150-156: Thank you for the helpful suggestion. We have revised this section to improve conciseness and clarity. The definitions and categorizations of obstetric variables have been summarized in a new table (S1 Table), and the text in the Methods section has been shortened accordingly.

5. Line 223: I suggest to change to – Maternal Care service status

Author response:

Line 206: Thank you for the suggestion. The subheading has been revised from “The coverage of maternal care” to “Maternal care service status” as recommended.

6. Line 237- I suggest to change to - Socio-demographic factors of the study subjects into Socio-demographic Characteristics of the study participants

Author response:

Line 220: Thank you for the suggestion. We have revised “Socio-demographic factors of study subjects” to “Socio-demographic characteristics of study participants.”

7. Table 1. use n= ....rather than N and format table as per PLOS One standards

Author response:

Thank you for the suggestion. We have revised Tables 1 and 2 to use “n” instead of “N” and reformatted the tables according to the PLOS One table style.

8. Line 247: the title is not representing the table you are presenting. You are showing the Cross tabulation, please review similar paper and format accordingly. I shall suggest to take consultation with an experienced statistician.

Author response:

Thank you for the helpful comment. We have revised the titles of Tables 1 and 2 to better reflect the content of the tables. The new title of Table 1 is “Socio-demographic characteristics of women according to completion of the maternal continuum-of-care.” The new title of Table 2 is “Obstetric characteristics of women according to completion of the maternal continuum-of-care.” The tables present cross-tabulations of socio-demographic variables by continuum-of-care completion status.

9. Line 266: the sub-heading should be something like- Association between socio-demographic factors and completion of maternal continuum of care rather than - Logistic regression analysis on completion of the maternal 267 continuum-of-care

Author response:

Lines 250-251: Thank you for the helpful suggestion. We have revised the subheading to better reflect the content of the analysis. The new subheading is “Factors associated with completion of the maternal continuum-of-care,” because not only socio-demographic factors but also obstetric variables were included.

10. Table 3. is perfect but format for clarity and we dont need P value here. In the footnote highlight which aOR or cOR is significant based on OR

Author response:

Thank you for your valuable comment. In Table 3, P-values are presented together with crude odds ratios (cOR) and adjusted odds ratios (aOR) to provide full statistical information. Statistical significance was defined as P < 0.05 and this has been clearly stated in the Methods section. In addition, statistically significant P-values are already highlighted in bold in the table, which allows readers to easily identify significant associations. Therefore, we believe that the current format provides clear and complete statistical reporting. For these reasons, we have retained the P-values in Table 3.

11. Discussion: First paragraph overall summarize your study findings. I suggest to rewrite your whole section analytically.

Author response:

Lines 282-288: Thank you for your valuable comment. We agree that the first paragraph of the Discussion should clearly summarize the key findings of the study. Therefore, we have revised the first paragraph to more clearly present the main results, including the overall coverage of the maternal continuum-of-care, the stage at which discontinuation most commonly occurred, and the factors associated with completion of the continuum-of-care. We have revised the first paragraph as follows, “This study examined the coverage and determinants of completion of the maternal continuum-of-care in Lao PDR using nationally representative LSIS III data. The findings revealed that the overall coverage of the maternal continuum-of-care was low (3.3%). Discontinuation of care most commonly occurred at the PNC stage. Factors associated with completion of the continuum-of-care included a higher wealth index, having the last child as male, and receiving a maternal health check before discharge. In contrast, Hmong-Mien women were less likely to complete the continuum-of-care than Lao-Tai women.”

12. Your discussion is too long and loosely written, I suggest to rewrite analytically and tighten the results.

Author response:

Thank you for your valuable comments. In response to your suggestion, we have substantially revised the Discussion section to improve clarity and analytical structure. The section has been shortened and reorganized to focus more clearly on the key findings and their interpretation. Specifically, we streamlined several paragraphs to avoid repetition and tightened the discussion of policy implications.

13. Where is your strength and limitation of the study, please include these too.

Author response:

Lines 351-356: We have added a new paragraph describing the strengths of the study and combined the strengths and limitations into a dedicated subsection in the Discussion.

14. Conclusion: Rework on conclusion, your writing need to tighten. Conclude based on the information.

Author response:

Lines 380-387: Thank you for your valuable comment. We have revised the Conclusion section to make it more concise and focused on the key findings of the study. The revised conclusion summarizes the main results and highlights their implications for improving maternal continuum-of-care in Lao PDR, while removing detailed policy descriptions to improve clarity and brevity.

---

## [Editor Report · Decision Letter 2]

9 Mar 2026

The coverage of maternal continuum-of-care and associated factors in the Lao People’s Democratic Republic: A population-based cross-sectional study

PONE-D-25-57622R2

Dear Dr. Yamamoto,

We’re pleased to inform you that your manuscript has been judged scientifically suitable for publication and will be formally accepted for publication once it meets all outstanding technical requirements.

Kind regards,

Kanchan Thapa, MPH, MPhil

Academic Editor

PLOS One
---

## [Editor Report · Acceptance letter]

PONE-D-25-57622R2

PLOS One

Dear Dr. Yamamoto,

I'm pleased to inform you that your manuscript has been deemed suitable for publication in PLOS One. Congratulations! Your manuscript is now being handed over to our production team.

Kind regards,

on behalf of

Mr. Kanchan Thapa

Academic Editor

PLOS One